# Techno-Functional Role of Exopolysaccharides in Cereal-Based, Yogurt-Like Beverages

**Valery Ripari**

Bioenologia 2.0, via G. Verdi, 32, 31046 Oderzo, Italy; rivyat@tiscali.it; Tel.: +39-0422815518

**Abstract:** This review describes the technical and functional role of exopolysaccharides (EPSs) in cereal-based, yogurt-like beverages. Many microorganisms produce EPSs as a strategy for growing, adhering to solid surfaces, and surviving under adverse conditions. In several food and beverages, EPSs play technical and functional roles. Therefore, EPSs can be isolated, purified, and added to the product, or appropriate bacteria can be employed as starter cultures to produce the EPSs in situ within the matrix. The exploitation of in situ production of EPSs is of particular interest to manufacturers of cereal-base beverages aiming to mimic dairy products. In this review, traditional and innovative or experimental cereal-based beverages, and in particular, yogurt-like beverages are described with a particular focus in lactic acid bacteria (LAB's) EPS production. The aim of this review is to present an overview of the current knowledge of exopolysaccharides produced by lactic acid bacteria, and their presence in cereal-based, yogurt-like beverages.

**Keywords:** cereal-based yogurt-like beverages; exopolysaccharides; LAB

## 1. Introduction

Since prehistoric times, man used cereals for food. Cereals are rich in vitamins, minerals, fibers, dietary carbohydrates, and proteins, even if they are lacking in essential amino acids. However, cereals contain anti-nutritional-factors (e.g., phytic acid, which is a mineral chelator), and some nutrients that are poorly digested [1]. Fermentation can improve nutritional value [2], and functional and sensorial properties [3] of cereals. A larger part of cereal products in the world is fermented to achieve different kind of beverages and dough. Fermentation is a traditional and low-cost method to enhance the food shelf-life naturally without additives or preservatives [4]. Traditional and innovative beverages made with cereals have received more attention in the last 30 years and the number of literature papers in this field have rapidly increased due to the consumer demand of functional foods and non-dairy beverages [5]. In particular, lactose intolerance, the level of cholesterol, awareness of consumer health, and the trend of vegetarianism and veganism are the principal causes of this new interest.

Usually a spontaneous consortium of lactic acid bacteria (LAB) and yeast strains ferments traditional cereal-based beverages or dough. Some of the LAB strains are able to produce exopolysaccharide (EPS) key compounds for non-dairy cereal beverages and for bakery products made with sourdough. EPSs are used as additive in food, beverages, pharmaceutical, cosmetics, biotechnology, agricultural, detergents, paint (e.g., stabilizer and emulsifier for thixotropic paints), paper (e.g., roll coating), textile (e.g., suspending agent for dies), and petroleum products. In cosmetics, EPSs are employed as a moisturizer due to their water retention capacity where they are usually used in creams and lotions [6].

In the agricultural field, some EPSs are employed as bio-surfactants, where EPS-producing bacteria are important in the rhizosphere of the crop plants for their roles in adhesion to soil, water retention, and for the nutrient flow across plant roots [6]. Xanthan gum is a bacterial EPS, and it is largely used in the petroleum industry in oil drilling, pipeline cleaning, and fracturing [6]. Dextran is a

flavorless homopolysaccharide composed of glucose subunits and is a GRAS (generally recognized as safe)-granted thickener used as an additive by the food industry.

However, not all industrial sectors benefit from EPS-producing bacteria. In the beverage industry, especially in brewing and wine fields, EPS-producing bacteria are considered a spoilage microorganism because they lead to a viscous and slimy product. In particular, EPSs lead to an alteration called "oilness" or "ropiness," which is characterized by a viscous, thick texture, and oily feel, and renders the products unpleasant because they alter the taste of the product [7]. This alteration has been described in wine, ciders, beers, and other fermented beverages. In particular, some LABs (*Lactobacillus*, *Oenococcus*, and *Pediococcus* strains) can produce C2-substituted (1 → 3)-β-glucans altering the fermented beverages with a "ropiness" texture [8]. A way to evaluate the microbial capacity to produce EPS is a plate test on an agar medium reach in sucrose [9], where after the growth of colonies, the microorganism able to produce EPS appears clearly slimy (Figure 1) or by assessing colonies with a sterile toothpick.

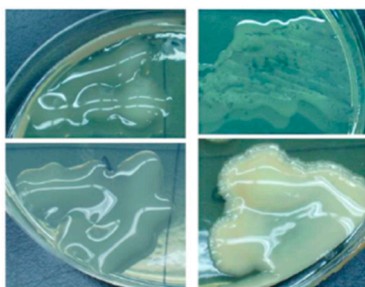

**Figure 1.** EPS production on an agar plate with the slimy appearance of the strains able to produce EPSs.

## 2. Type of Exopolysaccharides

EPSs can be divided into two groups (Figure 2), heteropolysaccharides (HePSs) and homopolysacharides (HoPSs). HePSs are synthesized intracellularly and are formed using more than one kind of monosaccharide; HoPSs are known as extracellular exopolysaccharides because are obtained via the action of an external enzyme, and are formed from only one kind of monosaccharide (subunits of glucose or fructose).

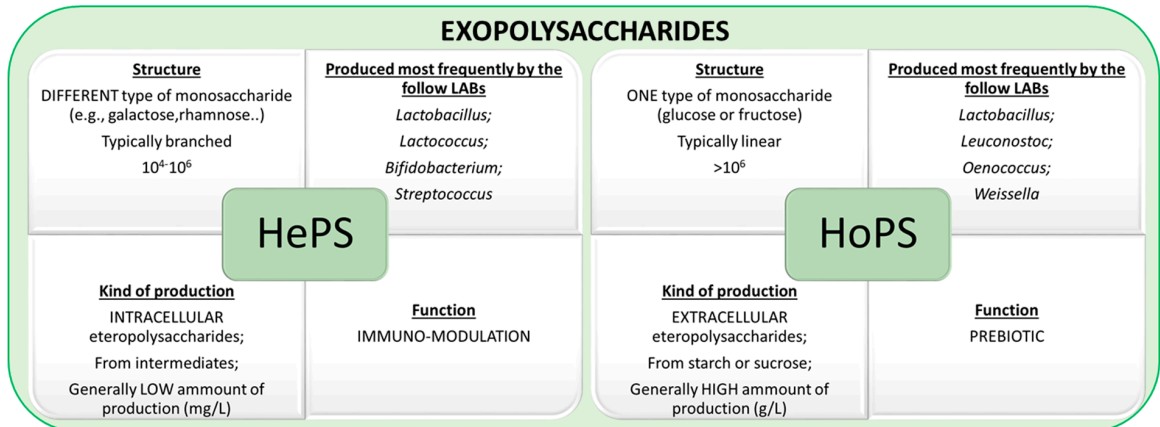

**Figure 2.** Schematic description of exopolysaccharides, including principal information about structure, production, and main function of EPSs.

In particular, HePSs can contain different monosaccharides in a range from two to eight and usually are glucose, rhamnose, or galactose, but can also include fructose, fucose, mannose, N-acetylglucosides, glucuronic acid, as well as contain phosphate or acetyl groups [10]. HePSs are typically branched and therefore they contain both α- and β- links. Their molecular mass ranges between $10^4 - 10^6$ Da [11]. The synthesis of HePSs involves four principal steps: (i) sugar transportation,

(ii) sugar nucleotide synthesis, (iii) repeating unit synthesis, and (iv) polymerization of the repeating units [12]. Some strains of *Lactobacillus*, *Lactococcus*, *Streptococcus*, and *Bifidobacterium* spp. can produce HePSs. In particular, HePS are produced via mesophilic and thermophilic LABs. The major mesophilic EPS-producer LABs are *Lactococcus lactis* subsp. *lactis*, *Lactobacillus rhamnosus*, *Lactobacillus sakei*, and *Lactobacillus casei*. Indeed, *Lactobacillus delbrueckii* subsp. *bulgaricus*, *Lactobacillus acidophilus*, *Lactobacillus helveticus*, and *Streptococcus thermophilus* are the major representatives of thermophilic LABs that are able to produce HePS.

HoPSs are named glucans (e.g., dextran) if they are composed of glucose; fructans (e.g., levan) if they are composed of fructose; or polygalactan if they are obtained from galactose. HoPSs are composed extracellularly from sucrose via glycansucrase or levansucrase and are generally produced in a high amount [13,14]. Depending on the linkage type and the position of the carbon involved in the bond, HoPS LABs can be sub-classified (e.g., α- or β-glucan).

Some strains of *Lactobacillus*, *Leuconostoc*, *Streptococcus*, *Weissella*, and *Oenococcus* spp. can produce HoPSs (Figure 2). The HoPS average molecular mass is up to $10^6$ Da [11].

## 3. Microbial Exopolysaccharides

Different microorganisms (several bacteria, algae, and fungi) are able to synthetize EPSs. The EPSs physiological role is strictly influenced by ecological niches and the habitat of the producing strain [15]. The ability of microorganisms to produce EPSs is strain specific, is an ecological advantage, and is a response to selective environmental pressures (Figure 3).

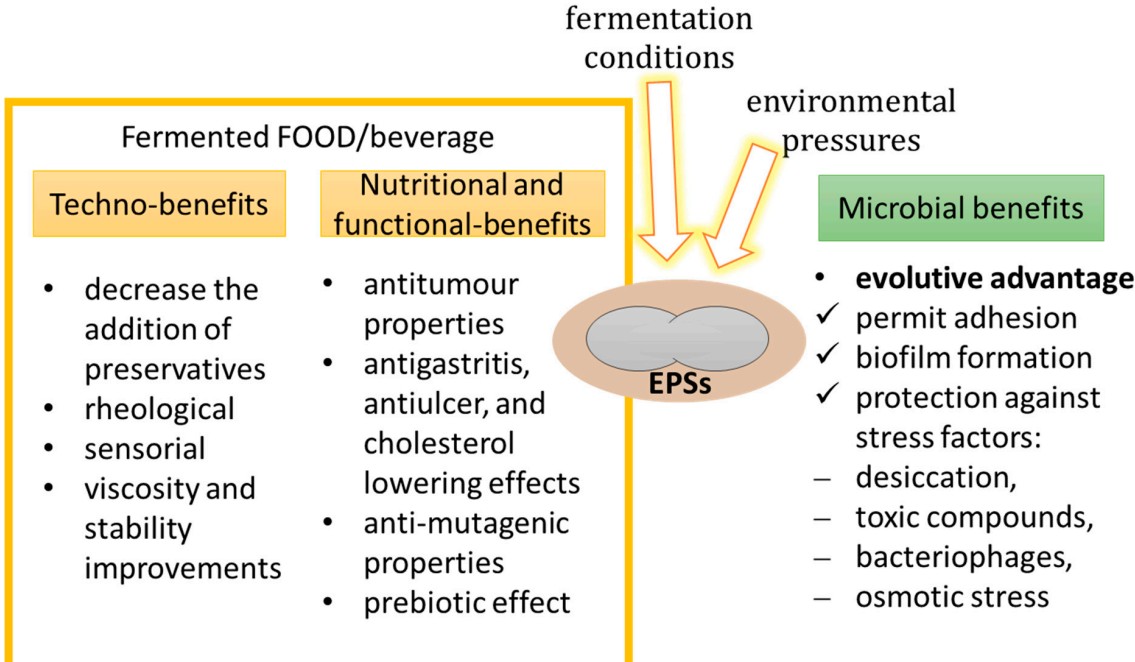

**Figure 3.** Microbial benefits due to EPS production for cell and key techno and functional roles of EPSs in food or beverages.

The EPS production process represents a carbon investment for the microbial cell; up to 70% of the total energy reserve of the cell is consumed for the EPS-making process [16]. Nevertheless, the benefits related to EPSs are higher than the costs. In particular, EPSs defend the cell against several types of stress. EPSs form a layer surrounding the microbial cell, and in this way, EPSs can protect cells against a change in temperature or in osmolarity, and against toxins and antibiotics. Indeed, several extremophiles are capable of producing EPSs, where this feature enables them to survive under the

otherwise lethal effect of extreme habitats [17]. EPS formation is thus necessary to the life of such microorganisms [18].

EPSs also play an important role in the mechanism of adhesion to other organisms or to the surface. EPS facilitate colonization through their adhesion to surfaces (e.g., adhesion to gut mucosa), and in immunomodulation [19]. Moreover, EPSs are important in bacterial biofilms [20]. Indeed, the EPSs confer structural integrity to the biofilms. It is been demonstrated that biofilm-forming microorganisms are 1000 times more resistant to antibacterial compounds such as surface-active compounds, antibiotics, and sanitizers [21].

Ruas-Madiedo et al. [22] demonstrated that EPSs produced by *Lactobacillus* and *Bifidobacterium* eliminated the cytotoxic effect of bacterial toxins on eukaryotic cells in vitro.

It is important to underline that the quantity and quality of EPSs produced is strictly dependent on the sugars available, on the presence of micronutrients that can act as enzymes cofactors (e.g., minerals), and fermentation conditions (e.g., temperature and time; [23,24]). Joshi and Koijam [25] studied the EPS production of *Leuconostoc lactis* isolated from Kyiad pyrsi (a rice-beer; North-East India beverage). They found that the carbohydrate source, temperature, and pH strictly influenced EPS production. The optimal conditions for EPS production were pH 6.5, 28 °C, and using sucrose as the carbon source.

## 4. EPSs Technical Roles

In general, EPSs, thanks to their physical properties, can be employed in food for several technical roles (Figure 3) such as: (i) influencing viscosity and rheology; (ii) improve texture, sensory properties, mouth feel, and freeze–thaw stability; (iii) softeners; (iv) suspending compounds; (v) dietary fibers; (vi) coating agents; (vii) salad dressings; (viii) frozen-food icing; and (ix) moisturizing agents. Due to their physical properties, EPSs can be added in food as additives, or foods can be fermented with microbial EPS-producing strains. EPSs interact with the water molecules and control the rheological properties and physical stability of foods. They are soluble in water and form a gel as the viscosity of food/beverage augments [12]. Some examples are as follows.

In the backing process, EPS-producing LAB strains may affect the technological properties of dough and bread regarding (i) water retention, (ii) rheological properties, (iii) stability while frozen, (iv) loaf volume, and (v) staling [26]. The major benefits of bread made with sourdough where EPS-producing LABs are present are flavor, texture, and shelf-life [27,28]. Ripari et al. [9] found that most of the 35 sourdoughs analyzed contained at least one strain of EPS-producer from sucrose, and EPS production seems to be an important feature in the sourdough microbial consortium.

In dairy products, EPSs can improve the rheological properties by influencing viscosity, firmness, syneresis, and sensory properties [29]. The major sensory traits for dairy products affecting consumer preference is firmness and creaminess. EPSs may act as texturizers by increasing the viscosity of the final product and as stabilizers by binding water and interacting with the other milk constituents, such as proteins and micelles, to fortify the firmness of the casein network. Therefore, EPSs can decrease the synthesis of harmful by-products and improve stability. In yogurts, EPS-producing bacterial strains lead to a higher viscosity and less phase separation. Han et al. [30] selected a *Streptococcus thermophilus* zlw TM11 strain from among 19 strains because it produced the highest EPS content (380 mg/L) and viscosity (7716 mpa/s) in fermented milk. Then, they combined this strain with a *Lactobacillus delbuecki* one to produce a yogurt, and they compared the results with yogurt obtained from commercial strains. Their combination (called SH-1) led to a better texture of yogurt and lower whey separation [30]. EPSs also play an important role in yogurt or milk fermentation made using low fat milk. Milk fat contributes to the flavor, body, and texture of the products, while the use of low-fat milk leads to functional and textural defects. The use of EPS-producer strains can prevent the weak body and poor texture of low-fat dairy products [31].

Saint-Eve et al. [32] highlighted that an increment in viscosity in yogurt (due to an EPS increment) can eventually reduce diffusion and the release of volatile compounds within the food matrix.

Juvonen et al. [33] studied the impact of EPS-producing LABs on rheological, chemical, and sensory properties of pureed carrots. The formation of low-branched dextran was correlated with thickness, while the production of β-glucan was correlated with elasticity perception. *Weissella confusa* and *Leuconostoc lactis* strains (low-branched dextran-producers) produced pureed carrots with a thick texture and a pleasant odor and flavor.

It is important to highlight that the rheological properties of EPSs depend on their concentration, composition, size, structure, and charge [34]. Branched bacterial β-glucan promote viscosity even at low concentrations; this is due to their linear rod-like structure [8]. Moreover, soluble β-linked EPSs with regular structures lead to gelation.

Otherwise, dextrans that are composed of α-linkages, both single unit and elongated branches, form compact ramified structures, and therefore tend to effect viscosity only at high concentrations [35]. The rheological properties of dextrans are influenced by their structural heterogeneity and solubility.

EPSs can maintain a high level of water-soluble active compounds from cereals due to their high water-binding ability, which results in increased water retention. EPSs can also interact with structure-forming components present in the medium. Therefore, EPSs can improve the metabolic absorption of certain minerals (e.g., calcium, magnesium, and iron).

## 5. EPSs Functional Roles

EPSs can positively affect gut health. EPSs are a non-digestible food fraction, and for this reason are prebiotic and impart beneficial effects on the human gastrointestinal tract. Salazar et al. [36] showed that EPSs synthesized by intestinal *Bifidobacteria* can be fermented by microorganisms in the human gut, and can therefore modify the interactions between gut microbiota. Therefore, EPSs are suggested to possess antitumor [37,38], antiulcer [39], and immunomodulatory [40] properties, and are proposed to decrease blood cholesterol values (Figure 3) [41,42].

Tok and Aslim [42] showed that some strains of *L. delbrueckii* subsp. *bulgaricus*, isolated from home-made yogurt and producing high amounts of EPSs, were able to absorb more cholesterol compared to low EPS-producing strains. They also compared free and immobilized cells and found that the immobilized cells were much more active in the adsorption of cholesterol. The EPSs way of reducing cholesterol levels is not yet completely understood. It is probably due to EPSs binding cholesterol and promoting its excretion, or indirectly increasing their conversion to bile through the stimulation of microbes with bile salt hydrolase activity concentration [43].

Rodríguez et al. [44] demonstrated that purified EPSs from *S. thermophilus* CRL 1190 can prevent chronic gastritis. The EPS-protein interaction seems to be responsible of the gastro-protective outcome.

The prebiotic role of EPS LABs has been shown by Korakli et al. [45]. The EPSs of LABs with proven prebiotic traits are HoPSs, probably because it is more difficult for the gut microbiota to degrade the HePSs' complex structure, limiting their prebiotic potential [46].

EPSs promote viscosity that increase the retention time of fermented product in the gastrointestinal tract. Increase of the retention time can help the colonization of probiotic bacteria. Therefore, EPSs can be metabolized using the colonic microorganisms to form short-chain fatty acids (e.g., acetate, propionate, and butyrate), and they can provide energy to epithelial cells and play an important role in colon cancer prevention [29].

Olano-Martin et al. [47] studied the in vitro prebiotic functionality of LAB EPSs. In particular, they simulated the transit through the large intestine using a batch-culture fermentation, and found that dextran and oligodextrans can stimulate a probiotic effect, and at the same time, they observed a decrease of undesirable bacteria (e.g., *bacteroides* and *clostridia*). Vanamu et al [48] studied the in vitro prebiotic and probiotic effects of PROEXO (a probiotic formulas) and analyzed its influence on simulated microbiota. The results showed that EPSs induce increases of LAB strains in the colon ascending segment and a significant decrease of *staphylococci* and *clostridia.*

Laiño et al. [49] explained the immunogenic properties of EPSs. In particular, it seems that the phosphate groups on the HePSs are important effectors of immune stimulation [50]. Tsuda et al. [51]

showed the anti-mutagenic properties of EPS-bound cells of a *Lactobacillus plantarum* strain where the mutagens (such as heterocyclic amines) were inactivated by EPS binding.

Few studies are focused on in vivo EPSs functional roles. Chabot et al. [52] found that EPSs from *L. rhamnosus* RW-9595M can stimulate TNF (tumor necrosis factor), IL-6 (interleukin 6), and IL-12 (interleukin 12) in human- and mouse-cultured immunocompetent cells. Looijesteijn et al. [53] investigated the resistance of *L. lactis* subsp. *cremoris* B40 EPS to digestion in vivo using rats fed with an EPS-containing diet over two weeks. They found that EPS was not digested through the gastrointestinal tract.

## 6. Cereal-Based Beverages

Several non-dairy cereal beverages are traditionally used in the world (Table 1); some examples include boza, amazake, bushera, chhang, chica, haria, mahewu, omegisool, pozol, bhaati jaanr, togwa, and kvass [54–66].

**Table 1.** Traditional cereal-based beverages.

| Traditional Beverages Based on Cereals | Microorganism | Cereals | Kind of Beverage | Origins | References |
|---|---|---|---|---|---|
| amazake | *Aspergillus* sp. | rice | sweet fermented rice drink | Japanese | [54] |
| bhaati jaanr | LABs; yeast (*Saccharomycopsis fibuligera, Rhizophus* sp.) | rice | staple food beverage | Nepal, India, Bhutan | [55] |
| boza | LABs (*Lactobacillus* sp., *Leuconostoc* sp., *Lactococcus* sp., *Pediococcus* sp.); yeast (*Saccharomyces cerevisiae, Candida* sp., *Geotrichum* sp.) | wheat, rye, millet, maize | sweet colloid beverage | Bulgaria, Albania, Turkey, and Romania | [56] |
| bushera | LAB (species of *Lactobacillus, Lactococcus, Leuconostoc, Enterococcus, Streptococcus, Weissella*) | sorghum and Millet | non-alcholic drink | Uganda | [57] |
| chhang | LAB ($1.7 \times 10^4$ such as *Lactobacillus* sp.) and yeast ($3.5 \times 10^4$ such as *Saccharomyces* sp.) | rice, barley, or millet | alcholic drink | Nepal, Tibet | [58] |
| Kvass | LAB (*L.casei, Leuc.mesenteroides*); yeast (*S. cerevisiae*) | rye | fermented, non-alcholic drink | Russia | [59] |
| haria | microbial consortia (LAB, *Bifidobacterium*, and yeast) | rice | beverage | India | [60] |
| mangisi | LAB ($9.03 \times 10^{10}$ CFU/mL); yeast and mould ($1.1 \times 10^7$ CFU/mL) | millet | sweet, sour, non-alcholic drink | Zimbabwe | [61] |
| Marcha | LABs (*L. plantarum*); yeast | rice | fermented, non-alcholic drink | India | [62] |
| ogi (or koko) | LABs (such as *L. plantarum, L. pantheris, L. vaccinostercus*); yeast (such as *Candida krusei, Clavispora, S. cerevisiae, Rhodotorula* sp.) and mould (such as *Aspergillus* sp., *Penicillium* sp.) | maize, sorghum, millet, wheat | non-alcholic drink; porridge | Nigeria, Ghana | [63] |
| omegisool | LAB (*Lactobacillus* sp., *Pediococcus* sp.) | millet | alcholic beverage | Korea | [64] |
| togwa | LABs (*Lactobacillus* sp., *Pediococcus* sp.); yeast (*Candida* sp., *Issatchenkia orientalis, Saccharomyces cerevisiae*) | maize, millet | non-alcholic drink | Africa | [65] |

Cereals used as a substrate are fermented using a microbial consortium selected spontaneously, similarly to what occurs in the process to obtain sourdough [67]. Usually LABs, yeast, and acetic acid bacteria (AAB) can be present in this kind of consortium. Moreover, innovative/experimental non-dairy beverages have been studied (Table 2). The common cereals used for this type of beverage are rice, millet, sorghum, maize, and wheat [68]. Other cereals, such as oat or spelt [69], and also pseudocereals (e.g., quinoa and amaranth; [70]) have been employed to obtain innovative, functional, and probiotic beverages.

Several LAB strains involved in the fermentation of Ogi (sorghum-based) were able to produce EPSs [71]. Sawadogo-Lingani et al. [63] reported that up to 89% of 264 strains of *L. fermentum* isolated from spontaneous fermentation of two West African sorghum beers, Dolo and Pito, were EPS producers. The thicker texture of Dolo, a characteristic conferred from the presence of EPSs, is important for product quality and for its sensorial features.

**Table 2.** Experimental cereal/pseudocereal-based, yogurt-like beverages with a focus on EPS-producer strains if present.

| LAB strain | EPS | Substrate | Reference |
|---|---|---|---|
| *L. lactis ARH74* | EPS-producer strain | quinoa | [69] |
| *L. delbrueckii* subsp. *bulgaricus* -NCFB 2772 | EPS-producer strain | oat | [72] |
| *L. plantarum*CCM 7039 *and B. longum CCM 4990* | unknown | rice | [73] |
| *L. plantarum* Lp90 | EPS-producer strain | oat | [74] |
| *P.damnosus* | eps | oat | [75] |
| *P.damnosus 2.6* | β-glucan | oat based medium | [76] |
| *L.brevis G-77* | α- and β-glucan | | |
| *W. confusa DSM 2019* | dextran | quinoa | [77] |
| *W. cibaria* WC4 | EPS-producer strain | emmer flour | [78] |
| *L. rhamnosus* GG | unknown | buckwheat | [79] |

Heperkan et al. [56] identified and evaluated some technological properties of thirteen LABs isolated from Boza for screening them and chose strains to use as adjunct cultures in Boza. All strains, except *Streptococcus macedonicus* (A15), produced EPSs. *Leuconostoc citreum* (E55) and *Lactococcus lactis* (A47) were the highest EPS-producing strains (2.39 and 1.98 g/L of EPSs, respectively). Boza is a highly viscous beverage, but gel formation is not desired, and for this reason, LAB strains with low-EPS production capability were selected from the authors. EPS-producing LABs are able to colonize cereal beverages, and generally, they are part of the spontaneous micro-consortium that is naturally selected. Identification of the microbial population of traditional cereal beverages can demonstrate that EPS-producer LABs found in cereals are a great substrate for growth. In these traditional beverages, yeasts are usually present. It could be interesting to undertake new experiments to understand how yeast can influence the LAB EPSs production and bioavailability.

## 7. EPS-Producing LABs in Experimental Cereal-Based, Yogurt-Like Beverages

EPS-producing LABs significantly contribute to texture, mouth feel, taste perception, and stability of cereal-based, yogurt-like beverages.

Grobben et al. [72] studied *L. delbrueckii* subsp. *bulgaricus* NCFB 2772; in particular, they found that NCFB 2772 increased the viscosity of an oat-based medium added with glucose as a supplementary carbon source. To improve EPS production, it required a longer incubation time and a lower temperature.

Magala et al. [73] employed rice flour for fermented beverages using *L. plantarum* CCM 7039, *Bifidobacterium longum* CCM 4990, and *Lactobacillus brevis* CCM 1815. The highest values of sensory parameters were observed in the sample fermented using *L. plantarum* and in a sample with a mixed culture of *L. plantarum* and *B. longum*. They analyzed the viscosity of the product and found that the viscosity of samples decreased significantly after one day of fermentation.

Russo et al. [74] found that the EPS-producer *L. plantarum* Lp90 (Table 2) seems to have a positive impact on the rheological features of a new oat-based product, although it was apparently lost during the storage.

Mårtensson et al. [80] selected LAB strains that are able to produce exopolysaccharides (EPS) in order to obtain an oat-based fermented product.

It is important to underline that the amount of EPSs can prevent the physical instability and the phase separation of the final beverage [74]. Mårtensson et al. [75] employed *Pediococcus damnosus* EPS producer strain in combination with the common yogurt starter to produce oat-based, yogurt-like beverages. A sensory evaluation of the beverages show that there were no differences between this and a dairy equivalent product.

*L. brevis* G-77 and *P. damnosus* 2.6 strains producing β-/α-glucan (Table 2) have been applied to produce fermented oat-based drinks characterized by an elastic texture and increased viscosity [76].

In Lorusso et al. [77], a quinoa-milk fermented beverage was obtained using different LAB starters: a probiotic *Lactobacillus rhamnosus* SP1; an EPS-producing *Weissella confusa* DSM 20194; and *L. plantarum* T6B10, which is a strain isolated from quinoa (Table 2). During fermentation, the EPSs (dextran type) synthesized by *W. confusa* DSM 20194 lead to a viscosity and water-holding capacity (WHC) increment, and the formation of a stable EPS-protein network with a consequent improving of textural characteristics of the beverage.

Ludena Urquizo et al. [69], employed three LAB strains (*L. plantarum* Q823, *Lactobacillus casei* Q11, and *Lactococcus lactis* ARH74, which is an exopolysaccharide producer) for the fermentation of a quinoa-based fermented beverage. After fermentation, the viscosity was reduced. *L. lactis* ARH74 was lost during the storage time; on the contrary, *L. plantarum* Q823 and *L. casei* Q11 were detected after a 28-day storage period. Unfortunately, no data about the *L. plantarum* Q823 and *L. casei* Q11 capacity to produce EPS were produced.

An experimental emmer (*Triticum dicoccum*) flour-based, yogurt-like beverage was produced using an *L. plantarum* strain isolated from emmer flour that was previously selected [78]. The beverage texture was improved by employing *Weissella cibaria* WC4, an EPS-producer strain [78].

Coda et al. [81] employed multicereals (rice, barley, emmer, and oat), soy flour, and red grape must for making an original yogurt-like beverage. In particular, two *L. plantarum* strains were used in combination for the fermentation, but EPS production was not analyzed. Five beverages were obtained and investigated for technological and sensorial properties. During storage, all samples showed a decrease in viscosity, probably because of the post-acidification process [81].

EPSs are dietary fiber, and their presence in cereal-based yogurt-like beverages lead to a better bioavailability of dietary fiber in the product.

The viscosity developed in the gastro-intestinal tract seems to be an important and critical variable for the physiological effects of dietary fibers. Cereal processing improves the bioavailability or bioactivity of fiber (e.g., b-glucan). The viscosity of an aqueous oat-gum solution improved post-prandial glucose and insulin responses in humans [82].

## 8. Conclusions

To confirm that EPSs are always of greater interest, several patents of EPS-producer LAB strains have recently been obtained [79,83].

In conclusion, EPSs are important in cereal-based, yogurt-like beverages because of their technological role in improving stability, rheological properties, the texture of products, the control

of flavor release, increased shelf-life, and substituted fat and protein as described in Williams and Phillips [34].

Several studies demonstrate the possibility of developing cereal-based, yogurt-like beverages. The presence of a strain of EPS producers can improve the technical and functional quality of the beverages, leading to a drink without phase separation, with low pH, and that is stable during the storage period. These kinds of products could be a good source of protein, fiber, vitamins, and minerals, making them important not only for the coeliac and/or lactose-intolerant population, but also a new alternative for all customers. Moreover, many studies suggest that this kind of beverages might support the growth and viability of probiotic LABs (e.g., References [84,85]).

**Funding:** This research received no external funding.

**Conflicts of Interest:** The author declares no conflict of interest.

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
