# Peer review of "Techno-Functional Role of Exopolysaccharides in Cereal-Based, Yogurt-Like Beverages"

_beverages, doi:10.3390/beverages5010016_

Reviewer 1 Report

The present review is interesting and  correctly, clearly and didactically written. However, I do not understand the importance of the tab. 1, considering that there are listed beverages based on cereals, but not yougurt-like and that they are characterized by a microbial consortium also made up of yeasts. I wonder: but in these beverages rich in EPS, how much does the presence of the yeasts affect? It could be interesting to understand how the interactions between yeast and bacteria affect the production of EPS. However, since the aim of the work is focused on bacteria producing eps, I suggest eliminating the table.

Furthermore, I do not understand the link among the table 1, text following with  the title of the review. Furthermore, I suggest to check the possible presence of patents on the production of EPS and to shorten the  text of paragraph 7 text that is only descriptive and not informative.

Author Response

The present review is interesting and  correctly, clearly and didactically written.

Point 1. However, I do not understand the importance of the tab. 1, considering that there are listed beverages based on cereals, but not yougurt-like and that they are characterized by a microbial consortium also made up of yeasts. I wonder: but in these beverages rich in EPS, how much does the presence of the yeasts affect? It could be interesting to understand how the interactions between yeast and bacteria affect the production of EPS. However, since the aim of the work is focused on bacteria producing eps, I suggest eliminating the table. Furthermore, I do not understand the link among the table 1, text following with  the title of the review.

Response 1: Thank you for your comments. Probably it is not clear the importance of table 1 and text following (paragraph 6). So I tried to change the text and I added an explanation of the paragraph, I hope that in this way you can consider the table adapted to the review. (in red)

Point 2. Furthermore, I suggest to check the possible presence of patents on the production of EPS and to shorten the  text of paragraph 7 text that is only descriptive and not informative

Response 2: ok, I added same patents at the end of the paper (lines 311-312) and I make changes also in paragraph 7 (in red).

Reviewer 2 Report

Dear Author,

After the review process, I have the following comments:

- at the section 5, you should  insert more data about EPS from LAB strains and human microbiota (for example, https://doi.org/10.1007/s13213-014-0947-3); 

- you should detail how EPS from cereal-based yogurt-like beverages express the biological activity in vivo vs. in vitro; 

- you should explain if the presence of EPS could maintain a high level of other active molecules from cereal; 

- which is the bioavailability of fiber from cereal in the beverages based on rheological properties of the product.

Best regards!

Author Response

Point 1. at the section 5, you should  insert more data about EPS from LAB strains and human microbiota (for example, https://doi.org/10.1007/s13213-014-0947-3); 

Response 1: ok, added. Please see lines 187-198 (in red)

Point 2. you should detail how EPS from cereal-based yogurt-like beverages express the biological activity in vivo vs. in vitro;

Response 2: ok, see section 5 lines 187-189/ 203-207. Unfortunately very few data are present in literature about in vivo studies on EPS biological activity (in red)

Point 3. you should explain if the presence of EPS could maintain a high level of other active molecules from cereal; 

Response 3: please, see paragraph 4 (lines 162-165, in red)

Point 4. which is the bioavailability of fiber from cereal in the beverages based on rheological properties of the product.

Response 4: please, see the end of paragraph 7 (lines 304-309, in red)